**Data Availability Statement:** All the data used for the study are available as supportive information.

# Incidence and predictors of mortality among neonates with respiratory distress syndrome admitted at West Oromia Referral Hospitals, Ethiopia, 2022. Multi-centred institution based retrospective follow-up study

**Bruck Tesfaye Legesse**[1]*, **Netsanet Melkamu Abera**[2], **Tewodros Getaneh Alemu**[3], **Kendalem Asmare Atalell**[3]

1 Department of Pediatrics and Neonatal Nursing, School of Nursing and Midwifery, Institutes of Health Science, Wallaga University, Nekemte, Ethiopia, 2 Department of Nursing, College of Medicine and Health Sciences, Dire Dawa University, Dire Dawa, Ethiopia, 3 Department Pediatrics and Child Health Nursing, School of Nursing, College of Medicine and Health Sciences, University of Gondar, Gondar, Ethiopia

* brucktesfaye143@gmail.com

## Abstract

### Introduction

Respiratory distress syndrome is the major cause of neonatal death. However, data on the mortality and predictors related to respiratory distress syndrome were scarce. Hence, this study aimed to assess the incidence and predictors of death among neonates admitted with respiratory distress syndrome in West Oromia Referral Hospitals, Ethiopia, 2022.

### Methods

A retrospective follow-up study was conducted among 406 neonates admitted with respiratory distress syndrome at five referral hospitals from, 1 January 2019 to, 31 December 2021 in West Oromia, Ethiopia. The data were collected using a structured checklist and participants were selected using simple random sampling technique. The data were entered into Epi data version 4.6.0.2 and exported to STATA version 14 for cleaning, coding and analysis. The Kaplan–Meier curve was used to estimate survival time. The Weibull regression model was fitted to identify the predictors of mortality and variables with a P-value < 0.05 was taken as significant predictors of mortality.

### Result

Four hundred six neonates with respiratory distress syndrome were included in the analysis. The overall incidence of neonatal mortality was 59.87/1000 neonates-days observations (95%CI: 51.1–70.2) with a proportion of 152 (37.44%) (95% CI: 32.7–42.2). The median time of follow-up was 11 days (95% CI: 10–23). Very low birthweight (AHR = 4.5, 95%CI: 2.0–10.9) and low birth weight (AHR = 3.1, 95%CI: 1.4–6.6), perinatal asphyxia (AHR = 2.7, 95%CI: 1.8–4), Chorioamnionitis (AHR = 2.2, 95%CI: 1.4–3.5) and multiple pregnancies

**Funding:** The author(s) received no specific funding for this work.

**Competing interests:** The authors declare that they have no competing interests.

**Abbreviations:** APH, Ante Partum Haemorrhage; AURH, Ambo University Referral Hospital; CDC, Center for Disease Control; CPAP, Continuous Positive Airway Pressure; EMDHS, Ethiopia Mini Demographic and Health Survey; HTN, Hypertension; MKRH, Mettu Karl Referral Hospital; MAS, Meconium Aspiration Syndrome; NICU, Neonatal Intensive Care Unit; NMR, Neonatal Mortality Rate; NRDS, Neonatal Respiratory Distress Syndrome; NSRH, Nekemte Specialized Referral Hospital; PPH, Post-Partum Haemorrhage; PROM, Premature Rupture of Membrane; RD, Respiratory Distress; SDG, Sustainable Developmental Goal; VQ, Ventilation to Perfusion Ratio; WHO, World Health Organization; WRH, Wollega Referral Hospital.

(AHR = 2.2, 95%CI: 1.4–3.4) increased the hazard of death, whereas, antenatal corticosteroid administration (AHR = 0.33, 95%CI: 0.2–0.7) was negatively associated with neonatal mortality.

## Conclusion and recommendation

High mortality rate of neonates with respiratory distress syndrome was observed. Chorioamnionitis, perinatal asphyxia, low birth weight and multiple pregnancies increase the, mortality hazard while administering antenatal corticosteroids decreases it. Thus, administering corticosteroids- before giving birth and special emphasis on children with Chorioaminoitis, asphyxia, low birth weight and multiple pregnancies is important for reducing neonatal mortality.

## Background

Neonatal respiratory distress syndrome or Hyaline membrane disease, is one of the common causes of respiratory distress in neonates [1, 2]. It is mainly caused by decreased surfactant synthesis [2], and manifested with apnoea, cyanosis, grunting, inspiratory stridor, nasal flaring, poor feeding, tachypnea, and retractions in the intercostal, subcostal, or suprasternal areas [3, 4]. In addition, results in atelectasis, a ventilation-perfusion (V/Q) imbalance, hypoventilation, hypoxemia, and hypercarbia [2, 3, 5, 6]. These signs and symptoms appear at or shortly after birth, and they worsen over the first 48 to 72 hours of neonatal life [3, 4, 7].

Respiratory distress syndrome also accounts for 30 to 40% of newborn hospital admissions with higher rate of mortality with an abrupt onset, severe illness, and rapid progression [8, 9]. Specifically, in the early newborn period, respiratory distress syndrome is the main cause of neonatal death and morbidity [6, 10–14]. In addition, neonatal respiratory distress syndrome is responsible for the highest number of neonates' admission into neonatal intensive care units with 34.7% in Iraq, 57% in India, 23.5% in Nepal, 26.2% in Nigeria, 67 and 52.9% in Egypt respectively [15–19]. While accounting for 12.8% in, 46.9% in, and 49.5% of neonatal death in Poland, Nigeria and Ethiopia respectively [19–21].

Neonatal respiratory distress syndrome often end-ups with morbidities like Broncho-Pulmonary dysplasia and prolonged hospital stay resulting in economic consequences [22, 23]. Respiratory distress syndrome is a recognized risk factor for intracerebral/intraventricular haemorrhage and periventricular leukomalacia which raises the possibility of cerebral palsy and lately epilepsy [24, 25]. Compared to neonates without respiratory distress syndrome, neonates with a history of the disease have a 60% higher risk of lower respiratory tract infection-related hospitalization in the infancy period [26]. Those new-borns with respiratory distress are more likely to die than neonates who do not have respiratory distress syndrome [10, 27].

Neonatal death from respiratory distress syndrome is more likely when certain risk factors are present and likes as prematurity [2, 28–30]. The other risk factor for mortality from respiratory distress syndrome is low birth weight [31, 32]. Neonatal sepsis is also a known risk factor that facilitates neonatal death from respiratory distress syndrome [30, 32].

Other common factors for death outcomes from RDS are maternal and other pregnancy-related complications like gestational diabetes, premature rupture of membranes, pre-eclampsia, perinatal infection, and abnormal placental attachments and detachments [30, 32–36].

Early screening, improved treatment for preterm and mothers, a surfactant for preterm, corticosteroid for mothers at risk for preterm birth, and high oxygen flow can help the neonates to survive death caused by RDS [2].

According to WHO 2019 report, Ethiopia ranks among the top five countries with the highest number of neonatal deaths when RDS is the major contributor [37]. Ethiopia has made little progress in lowering neonatal mortality, when the neonatal mortality rate is still 33/1000 by 2019 and only decreased to 27/1000 live births by 2020 [38]. However, the data on predictors of mortality from respiratory distress syndrome is scarce in Ethiopia.

Even though improving neonatal mortality is one of the indicators of the country's health and developmental measures, there is a paucity of research in this field. Meanwhile, RDS is acknowledged as the major contributor to neonatal mortality. In addition, the incidence of mortality among neonates with RDS and its predictors are not known in Ethiopia. Therefore, this study aimed to determine the incidence and predictors of mortality among neonates with RDS in West Oromia Referral hospitals, Ethiopia, 2022.

## Methods

### Study design and setting

Institution-based retrospective follow-up study design was employed. The study was conducted at five Referral Hospitals (i.e., Mettu Karl Referral Hospital (MKRH), Jimma University Specialized Hospital (JUSH), Wallagga University Referral Hospital (WURH), Nekemte Specialized Referral Hospital (NSRH) and Ambo University Referral Hospital (AURH)), which are in West Oromia. Each Hospitals provides a referral service for large proportion of populations in the West Oromia. MKRH gives referral service for two zones called Illuu Abba Boor and Bunnoo Bedellee, JUSH gives referral service for Jimma zone while WURH and NSRH give service for four Wallagga zones, East, West, Kellem and Horro Guduru Wallagga and AURH gives services for West Shewa zone. All the above mentioned Hospitals have an advanced NICU services, with a team of nurses, neonatal nurses and paediatricians.

### Study population

The source population for our study was all neonates with respiratory distress syndrome admitted in NICU ward at West Oromia Referral Hospitals whereas the study populations was all neonate diagnosed with RDS and admitted to NICU ward between January 2019 and December 2021 in the five Referral Hospitals, West Oromia.

Neonates with incomplete medical records (i.e., admission and discharge dates, gestational age, birth weight, and outcome variable) and neonates with major respiratory and cardiac system congenital malformations were excluded from the study.

### Sample size determination and sampling procedures

The sample size was determined by using single population proportion formula for the first objective and double population proportion formula for the second objective. The following statistical assumptions were considered for single population proportion; death rate = 50% (p = 0.5 and q = 0.5), 95% confidence level, and 5% margin of error.

$$\frac{(1.96)(1.96)(0.5)(0.5)}{0.05 \times 0.05} = \mathbf{384.16}.$$

By considering missed patient charts, 10% of the initial sample size was added and the final sample size was 423.

Sampling frame was prepared after collecting the medical registration number of patients from the registration books of each hospital. A proportional allocation was done for each Hospital and 423 neonates were selected by a simple random sampling technique using computer-generated random numbers.

## Study variables

The outcome variable of our study was survival status, dichotomized into two the event (death = 1) and censored = 0, if the neonates are alive until the end of the follow-up, discharged, and transferred out. We followed the neonates starting from the day of admission to the occurrence of an event or end of follow-up.

The explanatory variables of this study were 1) Socio-demographic variables such as age at admission, sex, place of birth, maternal age, and residence. 2)Antepartum characteristics of the mother such as ANC follow-up, parity, chronic hypertension, Diabetic Mellitus, pre/eclampsia, HIV, Rh incompatibility, APH, and Antenatal corticosteroid administration. 3) Intrapartum characteristics of the mother including mode of delivery, the onset of labour, labour duration, types of pregnancy, PROM, Chorioamnionitis, and obstructed labour. 4) Clinical characteristics of neonates are such as gestational age, birth weight, neonatal sepsis, perinatal asphyxia, MAS, Hypoglycaemia, Jaundice, Neonatal anaemia and Hypothermia.

## Operational definition

**Neonatal Respiratory Distress Syndrome:** diagnosed based on the presence of one or more of the following signs: an abnormal respiratory rate, expiratory grunting, nasal flaring, chest wall recessions with or without cyanosis [2].

**Event (death):** - Neonate died in the hospital who was admitted with RDS.

**Censored:** new-borns with RDS who did not develop the outcome of interest (death) until the end of the follow-up period or recovery from illness or discharged against medical advice.

**Time to death:** The time in days from admission to the development of the outcome variable (Death) within complete 28 days of follow-up time.

**Defaulted**- RDS cases that are signed (parents on behalf of their child) against treatment to leave the treatment before cure.

**Birth asphyxia**- Diagnosed based on the World Health Organization (WHO) definition of "the inability to initiate and maintain breathing at birth" [39].

**Chorioamnionitis**- Based on maternal fever accompanied by two of the following symptoms: maternal tachycardia, foetal tachycardia, uterine tenderness, foul odour of amniotic fluid, or maternal leucocytosis [40].

**PROM**- A gush of fluid passing through the birth canal as a result of the amniotic sac rupture before true labour begins [41].

**Obstructed labour**- When the foetus is unable to descend into the birth canal for mechanical reasons despite strong uterine contractions [42].

## Data collection tools and procedure

Data were extracted from patients' charts using a structured data extraction checklist adapted from different literatures [21, 22, 28–31, 43–46] by five BSc Nurses working at NICU wards in each respective hospital and supervised by the principal investigator and two supervisors. The medical registration number (MRN) was used to identify individual patient cards. Socio-demographic and other clinical data such as date of admission, discharge, and other neonatal and maternal clinical profiles were collected accordingly from the medical records of the neonates and the mothers.

## Data quality control

To ensure the quality of data, the data extraction checklist was pretested on 5% of the sampled charts at Wallgga University referral hospital. Data were collected by trained nurses having work experience in the NICU ward from respective hospitals. The data retrieval process was closely monitored by the principal investigator and supervisors. Besides, each data extraction tool was checked for its completeness immediately after completion by supervisors and the principal investigator.

## Data processing and analysis

The data were checked for consistency and completeness and entered to Epi-data version 4.6.0.2 and then exported to STATA version 14 for cleaning, coding, and analysis.

Descriptive statistics were carried out and data were summarized using tables, graphs and texts. The incidence rate of mortality was calculated for the entire follow-up by dividing the total number of cases by the total person-days of follow-up. The Nelson Aalen cumulative hazard was used to estimate the overall survival time. The Log-rank test was used to compare survival experiences of each categorical variables. Proportional hazard assumption was checked both graphically and using a Schoenfeld residual test which assesses the relationship between the scaled Schoenfeld residuals and time. Multicollinearity was checked between independent variables by a variance inflation factor with 1.39 mean VIF. The frailty model was considered and checked for the random effect. The log-likelihood, and Akaike Information Criteria (AIC) were applied to select the best-fitted model, and a model with a minimum AIC value was considered (i.e., Weibull regression model) as the best-fitted model (Table 5). The goodness of fit of the model was assessed by Cox-Snell residuals with the Nelson Aalen cumulative hazard function graph. Variables with $P < 0.25$ in the bi-variable analyses were candidate for the multivariable analysis. Variables having a p-value $\leq 0.05$ in the multivariable model were declared as significant predictors of the outcome of interest. Finally, the crude and adjusted hazard ratio (HR) with 95% CI was calculated and used as the measure of associations.

## Ethical determination

Ethical clearance was obtained from the ethical review committee of the school of nursing on behave of the ethical institutional review board of the University of Gondar (Ref.no SN/229/2014). Permission letter was granted from each Hospital's administrators. Confidentiality was maintained at all levels of the study. Data was held on a secured password-protected system. All the procedures were conducted based on the principles of the Helsinki declaration [47].

## Results

### Socio-demographic characteristics

A total of 406 medical records were included in the analysis. Nearly two-thirds, 249(61.33%) of neonates were males. About 291 (71.67%) of neonates were admitted within the first 24 hours of birth. The mean age of mothers was 27 (SD: ±6) years old, and 88.92% belonged to the age group of 20–34 years old and 222 (54.68%) of them were Urban residents (*Table 1*).

### Antepartum characteristics of the mother

Nearly half, 182(44.83%) of the participants were primipara, and majority of the participants, 345(84.98%) were attending ANC follow-up while only 61 (15.02%) mothers were taken antenatal corticosteroid (*Table 2*).

**Table 1. Socio-demographic characteristics of the mother and newborn with RDS from 2019 to 2021, Ethiopia, 2022 (N = 406).**

| Variables | Frequency | Percent |
|---|---|---|
| **Sex of the Newborn** | | |
| **Male** | 249 | 61.33 |
| **Female** | 157 | 38.67 |
| **Neonatal admission age (in hrs)** | | |
| **≤ 24** | 291 | 71.67 |
| **>24** | 115 | 28.33 |
| **Maternal age (in years)** | | |
| **15–19** | 10 | 2.46 |
| **20–24** | 102 | 25.12 |
| **25–29** | 171 | 42.12 |
| **30–34** | 88 | 21.67 |
| **≥ 35** | 35 | 8.62 |
| **Residence** | | |
| **Urban** | 222 | 54.68 |
| **Rural** | 184 | 45.32 |
| **Place of delivery** | | |
| **The same facility** | 289 | 71.18 |
| **Referred** | 104 | 25.62 |
| **Home** | 13 | 3.2 |

## Intrapartum obstetric factors of the mother

Almost two third, 272 (67%) of the respondents were delivered through spontaneous vaginal delivery while 386 (95.07%) of the mothers' started labour spontaneously. Seventy-nine 19.46% of the mothers were diagnosed with Chorioaminoitis when 56 (13.79%) mothers were multiple pregnancy (*Table 3*).

## Clinical characteristics of neonates

The mean gestational age of the newborn was 36 (±2.6 SD) weeks. Nearly two-third (63.05%) of the new-borns were preterm, and 42(10.34%) were belong to the very low birth weight group. About a quarter (24.38%) of the neonates were diagnosed with PNA alongside with RDS (*Table 4*).

## Overall neonatal outcome

From the follow-up outcome, 3.45% were lost to follow-up or left the hospitals without health professionals' permission or against medical advice while 37.4% were died (Fig 1).

## Proportional hazard assumption

Based on the proportional hazard assumption test using Schoenfeld residual, all of the covariates fulfilled the assumption, and the overall global model satisfies the proportional hazard assumption (0.5725).

## Model comparison

After the proportional hazard assumption was checked, both semi-parametric and parametric proportional hazard models were fitted to estimate the survival time and identify predictors of

**Table 2. Antepartum characteristics of the mother of the new-borns with RDS from 2019 to 2021, Ethiopia, 2022 (N = 406).**

| Variables | Frequency | Percent |
|---|---|---|
| **ANC** | | |
| **Yes** | 345 | 84.98 |
| **No** | 61 | 15.02 |
| **Parity** | | |
| **Primipara** | 182 | 44.83 |
| **Multipara** | 224 | 55.17 |
| **Chronic Hypertension** | | |
| **Yes** | 6 | 1.48 |
| **No** | 400 | 98.52 |
| **Pre/eclampsia** | | |
| **Yes** | 15 | 3.69 |
| **No** | 391 | 96.31 |
| **Diabetic Mellitus** | | |
| **Yes** | 14 | 3.45 |
| **No** | 392 | 96.55 |
| **APH** | | |
| **Yes** | 20 | 4.93 |
| **No** | 386 | 95.07 |
| **Antenatal corticosteroid** | | |
| **Yes** | 61 | 15.02 |
| **No** | 345 | 84.98 |
| **HIV** | | |
| **Reactive** | 14 | 3.45 |
| **Non-reactive** | 392 | 96.55 |
| **RH** | | |
| **Positive** | 330 | 81.28 |
| **Negative** | 76 | 18.72 |

ANC = Antenatal care, APH = Antepartum haemorrhage, HIV = Human Immune Virus, RH = Rhesus factor

mortality among neonates with respiratory distress syndrome. Models were compared graphically and statistically by using Akaike information criteria (AIC) and log-likelihood to select the most parsimonious models for the data set. Based on this, the Weibull regression with the AIC = 664.0967 model was the best-fitted model (Table 5) (Fig 2).

The baseline hazard of death was fitted with Weibull distribution with a shape parameter (P) greater than one (P = 1.213, 95%CI: 1.1–1.72) which indicates the hazard is constantly increasing. The shared frailty was estimated and theta to be significantly different from zero (theta: $7.81^{e-08}$) but this shows that the distribution of unmeasured variables is indifferent between the hospitals.

## Incidence of mortality

The total neonate-days observation for the entire follow-up time was 2539 person-days when 421 from WURH, 397 from NCSH, 439 from MKCSH, 763 from JUSH, and 519 from AURH with minimum and maximum follow-up times of 1 and 28 days respectively. The median follow-up time was 11 days (95% CI: 10–23). During the follow up 152 (37.44%) (95% CI: 32.7–42.2) of the neonates developed the event (died). The overall incidence of mortality was 59.87

**Table 3. Maternal intrapartum obstetric factors for new-borns with RDS from 2019 to 2021, Ethiopia, 2022 (N = 406).**

| Variables | Frequency | Percent |
|---|---|---|
| **Mode of delivery** | | |
| **SVD** | 272 | 67 |
| **CS** | 112 | 27.59 |
| **Instrumental assisted** | 22 | 5.42 |
| **Onset of labour** | | |
| **Spontaneous** | 386 | 95.07 |
| **Induced** | 20 | 4.93 |
| **Labour duration** | | |
| **≤18hrs** | 386 | 94.33 |
| **>18hrs** | 23 | 5.67 |
| **Types of pregnancy** | | |
| **Single** | 350 | 86.21 |
| **Multiple** | 56 | 13.79 |
| **PROM** | | |
| **Yes** | 68 | 16.75 |
| **No** | 338 | 83.25 |
| **Chorioaminoitis** | | |
| **Yes** | 79 | 19.46 |
| **No** | 327 | 80.56 |
| **Obstructed labour** | | |
| **Yes** | 68 | 16.75 |
| **No** | 338 | 83.25 |

SVD = spontaneous vaginal delivery, CS = caesarean section, PROM = Premature rupture of membrane

per 1000 neonate days observations (95%CI: 51.1–70.2). The incidence of death at the end of the first 24hrs, 3rd day, 7th day, 14th, and > 14 days was 86.2, 71.96, 50.8, 48.5, and 27.2 per 1000-neonate day's observation respectively.

The incidence of death was 59.38(40.12–87.88) per1000 neonates-days-(ND) for WURH, 65.49(44.59–96.18) per 1000 ND for NSRH, 56.94(38.48–84.28) per 1000 ND for MKCSH, 57.67(42.92-77-49) per 1000 ND for JUSH and 61.66(43.6–87.2) per 1000ND for AURH.

**Overall failure function (survivorship function).** The overall Kaplan-Meier failure function showed that the probability of death among neonates with RDS was increased during the follow-up period. During the first day of admission, a 9% probability of death was observed. The cumulative probability of death at the end of 5, 10, and 15 days was 0.32, 0.49, and 0.59 respectively and the least probability of death was observed at the end of 15 days of follow-up time (Fig 3).

## Predictors of mortality among neonates with respiratory distress syndrome

The test of equality for survival distribution of different categories of independent variables was performed with Kaplan-Meier curve (log-log plot) and the log-rank test. In general, the pattern that one group's survivorship function lying above another group showed that the upper curve had a better probability of survival compared to the lower curve. On the other hand, the probability of death was high among the lower groups described by the Kaplan-Meier survival curve. Furthermore, the log-rank test confirmed whether the observed difference seen on the KM graph was statistically significant or not.

**Table 4. Clinical characteristics of the newborn with RDS from 2019 to 2021, Ethiopia, 2022, (N = 406).**

| Variables | Frequency | Percent |
|---|---|---|
| **Gestational age** | | |
| **Preterm** | 256 | 63.05 |
| **Term** | 150 | 36.95 |
| **Birth weight (in grams)** | | |
| **< 1500** | 42 | 10.34 |
| **1500–2499** | 183 | 45.07 |
| **≥ 2500** | 181 | 44.58 |
| **Neonatal sepsis** | | |
| **Yes** | 145 | 35.71 |
| **No** | 261 | 64.29 |
| **PNA** | | |
| **Yes** | 99 | 24.38 |
| **No** | 307 | 75.62 |
| **MAS** | | |
| **Yes** | 40 | 9.85 |
| **No** | 366 | 90.15 |
| **Jaundice** | | |
| **Yes** | 32 | 7.88 |
| **No** | 374 | 92.12 |
| **Neonatal hypothermia** | | |
| **Yes** | 288 | 70.94 |
| **No** | 118 | 29.06 |
| **Neonatal Anaemia** | | |
| **Yes** | 141 | 34.73 |
| **No** | 265 | 65.27 |
| **Neonatal hypoglycaemia** | | |
| **Yes** | 38 | 9.36 |
| **No** | 368 | 90.64 |

PNA = perinatal asphyxia. MAS = Meconium aspiration syndrome.

In the current study, new-borns with very low and low birth weight have lower survival times compared with normal birth weight. Moreover, neonates without PNA have a more favourable survival probability than those neonates with PNA. This study also revealed that neonates born from mothers' without Chorioaminoitis had a better probability of survival compared with their counterparts. Furthermore, neonates born from mothers who had taken antenatal corticosteroids had a better probability of survival compared with neonates born from mothers who did not take antenatal corticosteroids during pregnancy while neonates born single had a better survival probability than those born multiple. These differences were statistically significant with a p-value < 0.001 in a log-rank test (Fig 4).

## Bi-variable and multi-variable Weibull regression analysis

In the bi-variable Weibull regression analysis, age of the mother, neonatal age at admission, residence, ANC follow-up, Antenatal corticosteroid, Chorioaminoitis, RH, sex, PROM, obstructed labour, parity, types of pregnancy, birth weight, gestational age, neonatal hypogly-caemia, neonatal anaemia, Neonatal hypothermia, PNA and mode of delivery were eligible to multi-variable analysis with P- value less than 0.25. However, in the multivariable Weibull

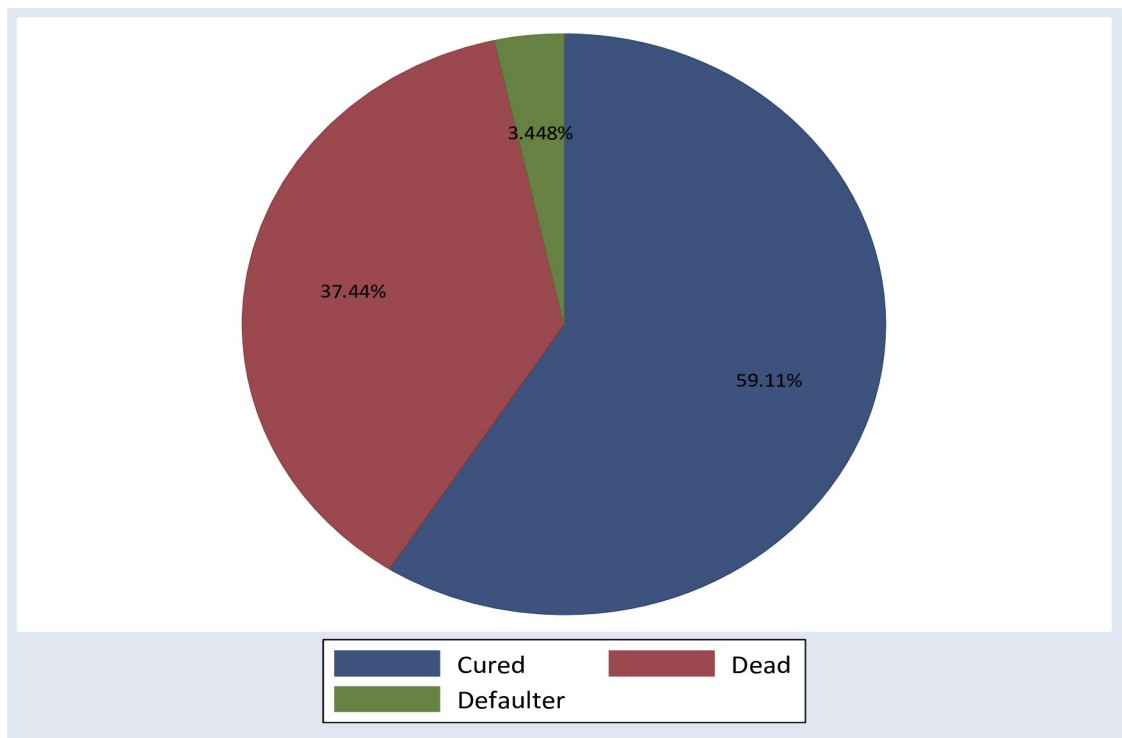

**Fig 1. Last outcome of neonates with respiratory distress syndrome from 2019 to 2021, Ethiopia, 2022, (N = 406).**

regression analysis, pregnancy type, antenatal corticosteroid, Chorioaminoitis, birth weight, and PNA were identified as the significant predictors of mortality among neonates with RDS at p-value less than 0.05 (Table 6).

According to our analysis, the hazard of death is two times among neonates with multiple birth as than singleton birth (**AHR: 2.2; 95%CI: 1.4–3.4**). The hazard of death is 67% less in mothers taking corticosteroids in antenatal periods as compared with their counter parts. (**AHR: 0.33; 95%CI: 0.2–0.7**). At any given time, the hazard of death in neonates born from mothers with Chorioaminoitis or infection during the third trimester were two times (**AHR = 2.2; 95%CI: 1.4–3.5**) compared to neonates born from mothers without Chorioaminoitis. The hazard of death was five (**AHR: 4.5; 95%CI: 2.0–10.9**) and three (**AHR: 3.1; CI: 1.4–6.6**) times for neonates being born with very low birth weight (<1500gm) and low birth weight (1500-2499gm) as compared to normal birth weight neonates. At any given time, the risk of death of neonates with RDS three times higher among neonates with perinatal asphyxia than neonates without PNA (**AHR: 2.7; 95 CI: 1.8–4**) (Table 6).

**Table 5. Summary of model comparison among the Cox proportional hazard model, parametric Regression models using AIC and LR criteria.**

| Model | Baseline Hazard | Log-likelihood | AIC |
|---|---|---|---|
| Cox | Unspecific | -735.2223 | 1518.445 |
| Weibull | Weibull | -306.0483 | **664.0967** |
| Exponential | Exponential | -310.1609 | 670.3218 |
| Gompertz | Gompertz | -310.1476 | 672.2951 |

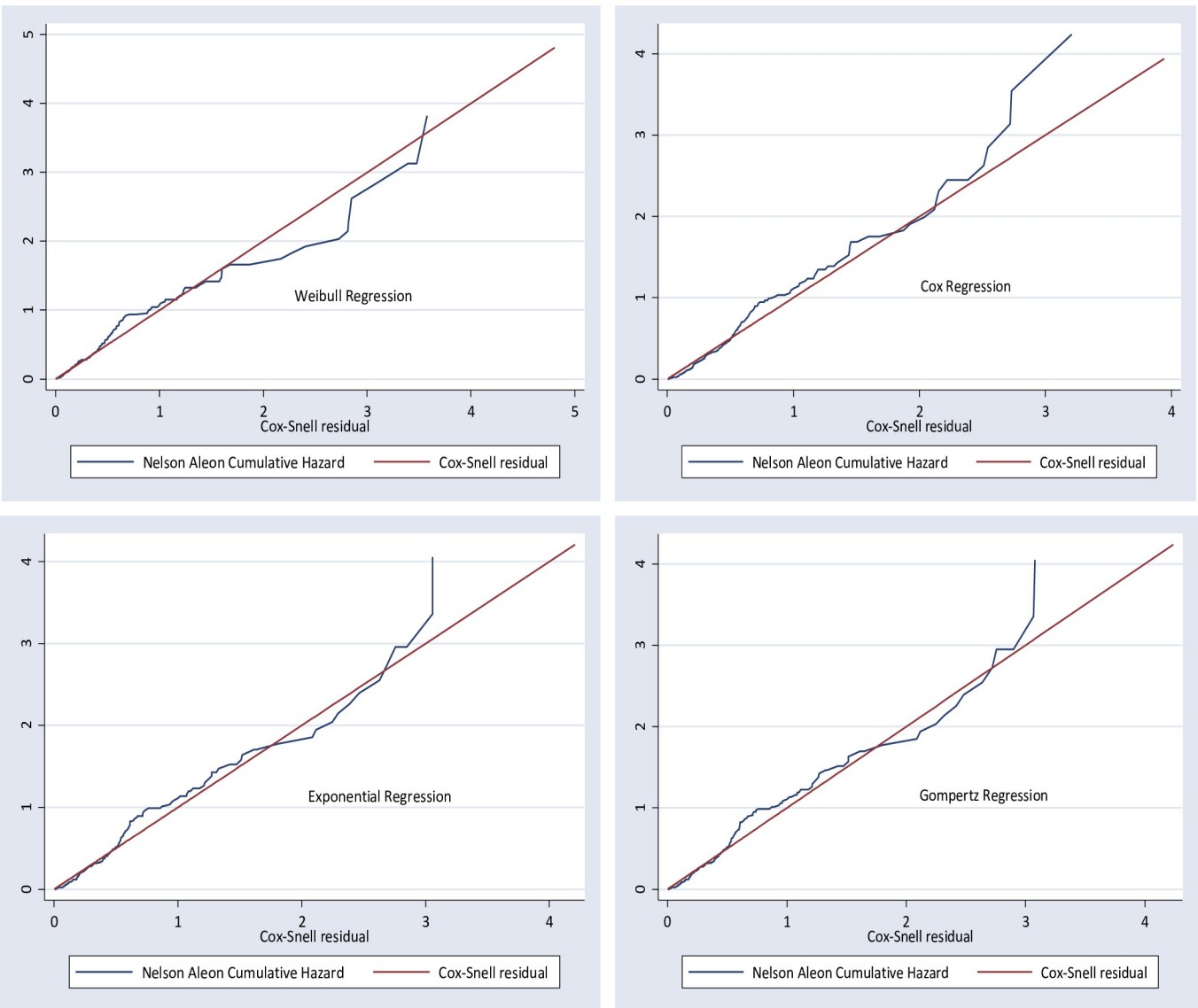

**Fig 2. Model comparison among Cox proportional hazard model and parametric models using Nelson Aalen cumulative hazard function graph, 2022.**

## Discussion

This study was aimed to investigate the mortality rate of neonates with respiratory distress syndrome and to identify the predictors of death. In this study 152 (37.44%) (95% CI: 32.7–42.2) neonates with the overall incidence of death was 59.87 (95% CI: 51.1–70.2) per 1000 neonate-day observations. The proportion of death in our study is in line with studies conducted at the University of Gondar Comprehensive Specialized Hospital, Ethiopia (32%) [48], and Bangladesh (36.5%) [31]. However, our finding was higher than studies conducted in China (8.06%) [36], Jeddah, Saudi Arabia (5.1%) [49] and Fiji [46]. The difference in the Fiji study might include that the source population for Fiji study was all live birth while this was on neonates with RDS. In addition, the study in Fiji doesn't incorporate some neonatal clinical comorbidities like PNA which is one of the risk factors for RDS. The definition of hypoglycaemia in the

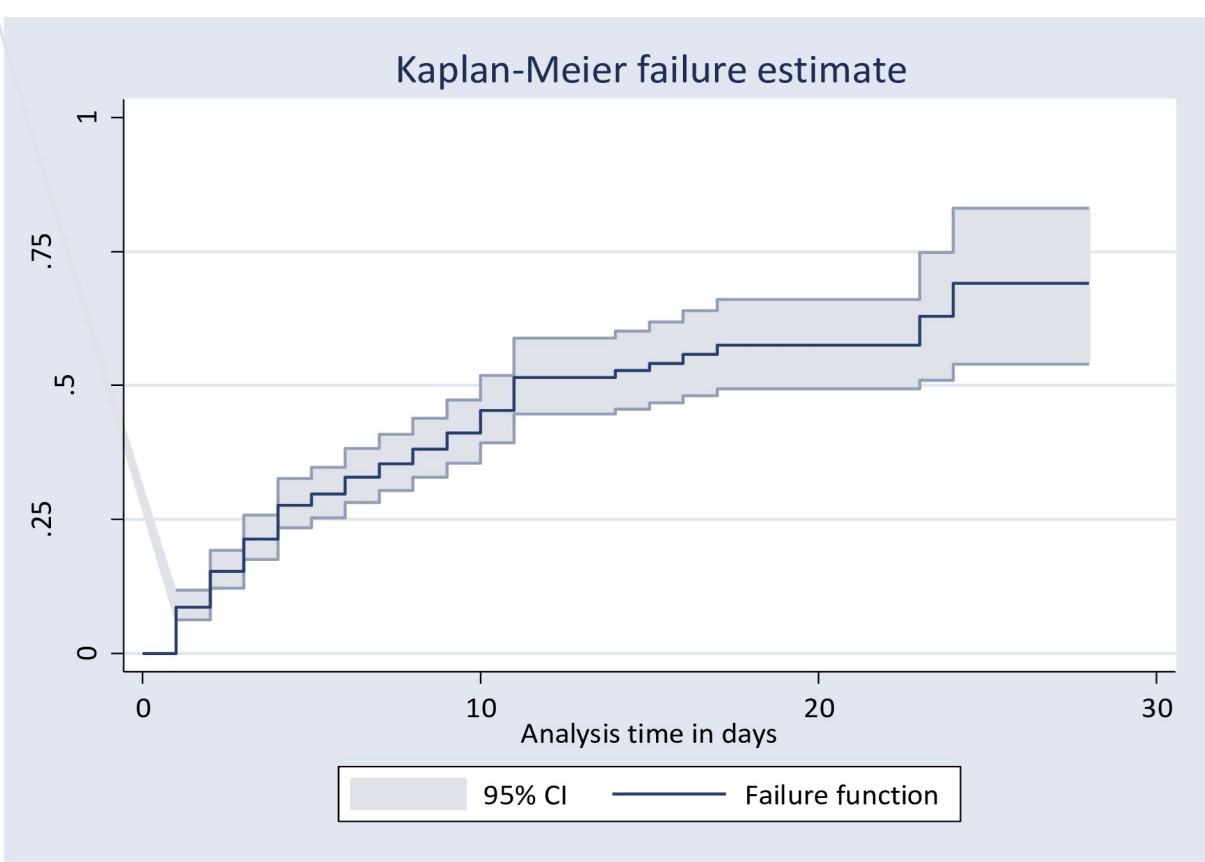

**Fig 3. Kaplan Meier failure function of death among neonates with neonatal respiratory syndrome from 2019 to 2021, Ethiopia, 2022 (N = 406).**

Fiji study was less than 56.8mg/dl while the current study is less than 40mg/dl, which might have impact on the mortality rate.

The possible reason for the discrepancy of results with China study might be due to the difference in NICU setup like using bubble CPAP for treatment, other advanced treatment technologies, surfactant administration-based treatment, and study design differences. Another possible reason for the difference between this study and the study in China is, that the study conducted in China was on term neonates only whereas the current study was conducted on all neonates irrespective of their gestational age [36].

The possible discrepancy between the current study and the study conducted in Saudi Arabia was conducted on term neonates only while the current study used all types of neonates as study unit, and the data were collected by research team whereas the current study was by data collectors to prevent researcher biases, and RBS, and PNA were not taken into consideration for study in Jeddah [49].

On the other hand the result in our study is lower than study conducted in Cameron (24.5%) [33], Kenya (72.3%) [45], Eritrea (48.1%) [50], and Northwest Ethiopia (49.5%) [21]. The possible justification for this difference may be the sample size and study setting. The study in Kenya was only conducted among 92 neonates and mono-centric, while the current study is multi-centre. The other possible reason for this difference might be the study in Kenya was only within preterm and low birth weight which facilitates neonatal death while the current study includes all birth weight groups [45].

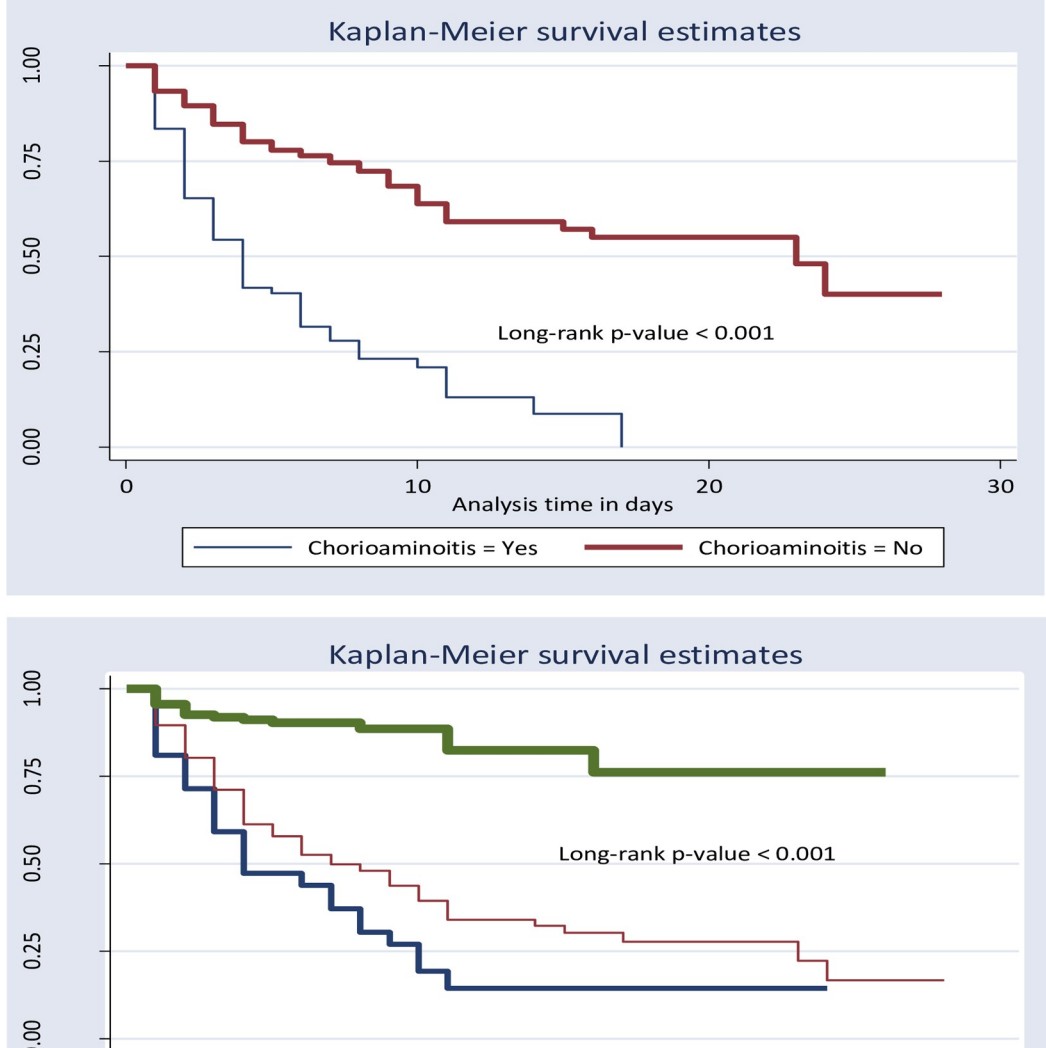

**Fig 4. Kaplan- Meir survival function for Chorioaminoitis and birth weight among neonates with respiratory distress syndrome from 2019 to 2021, Ethiopia, 2022, (N = 406).**

The mortality rate in each hospital ranged from 59.38(40.12–87.88) per1000 neonates-days-(ND) for WURH, 65.49(44.59–96.18) per 1000 ND for NSRH, 56.94(38.48–84.28) per 1000 ND for MKCSH, 57.67(42.92-77-49) per 1000 ND for JUSH and 61.66(43.6–87.2) per 1000ND for AURH. 56.94 deaths per 1000neonates days in Mettu Karl Comprehensive Specialized Hospital to 65.49 in Nekemte Specialized Referral Hospital. The possible reason for this disparity might be the paucity of highly skilled professionals such as paediatrician in NSRH unlike MKCSH which have senior paediatricians, residents and specialized neonatal nurses, which is integrated with the University.

**Table 6. Bivariable and multivariable Weibull regression analysis for predictors of death among neonates with RDS, from 2019 to 2021, Ethiopia (406), 2022.**

| Variables | Status | | CHR(95%CI) | AHR (95%CI) |
|---|---|---|---|---|
| | Event | Censored | | |
| **Maternal age (in years)** | | | | |
| **25–29** | 70 | 101 | 1 | 1 |
| **15–19** | 5 | 5 | 1.45(0.58–3.6) | 0.54(0.18–1.6) |
| **20–24** | 50 | 52 | 1.4(0.95–1.96) | 0.71(0.45–1.1) |
| **30–34** | 18 | 70 | 0.49(0.28–0.82) | 0.9(0.49–1.6) |
| **$\geq$ 35** | 9 | 26 | 0.71(0.36–1.43) | 0.59(0.26–1.3) |
| **Residence** | | | | |
| **Urban** | 66 | 156 | 1 | 1 |
| **Rural** | 86 | 98 | 1.62(1.2–2.23) | 1.2(0.82–1.73) |
| **Parity** | | | | |
| **Multipara** | 60 | 164 | 1 | 1 |
| **Primipara** | 92 | 90 | 2.2(1.57–3) | 1.3(0.8–2.1) |
| **ANC follow-up** | | | | |
| **Yes** | 113 | 232 | 1 | 1 |
| **No** | 39 | 22 | 2.6(1.8–3.8) | 1.5(0.9–2.32) |
| **Types of pregnancy** | | | | |
| **Single** | 112 | 238 | 1 | 1 |
| **Multiple** | 40 | 16 | 3.5(2.4–4.98) | **2.2(1.4–3.4)*** |
| **Antenatal Corticosteroids** | | | | |
| **Yes** | 9 | 52 | 0.326(0.16–0.6) | **0.33(0.2–0.7)** |
| **No** | 143 | 202 | 1 | 1 |
| **PROM** | | | | |
| **Yes** | 44 | 24 | 3(2.1–4.3) | 0.99(0.6–1.7) |
| **No** | 108 | 230 | 1 | 1 |
| **Chorioaminoitis** | | | | |
| **Yes** | 60 | 19 | 4.23(3–5.9) | **2.2(1.4–3.5)** |
| **No** | 92 | 235 | 1 | **1** |
| **Obstructed labour** | | | | |
| **Yes** | 36 | 32 | 1.96(1.35–2.9) | 1.4(0.89–2.1) |
| **No** | 116 | 222 | 1 | 1 |
| **Age (in hrs)** | | | | |
| **$\leq$ 24** | 120 | 171 | 1.6(1.1–2.35) | 1.3(0.8–.2.) |
| **>24** | 32 | 83 | 1 | 1 |
| **RH** | | | | |
| **Negative** | 22 | 54 | 0.75(0.48–1.18) | 0.85(0.56–1.4) |
| **Positive** | 130 | 200 | 1 | 1 |
| **Sex** | | | | |
| **Male** | 79 | 170 | 0.82(0.6–1.13) | 1.4(0.9–1.98) |
| **Female** | 73 | 84 | 1 | 1 |
| **Gestational age(weeks)** | | | | |
| **Preterm** | 132 | 124 | 3.96(2.5–6.33) | 1.3(0.58–2.97) |
| **Term** | 20 | 130 | 1 | 1 |
| **Hypoglycaemia** | | | | |
| **Yes** | 30 | 8 | 3.03(2.02–4.53) | 1.5(0.95–2.4) |
| **No** | 122 | 246 | 1 | 1 |
| **Neonatal Anaemia** | | | | |

(*Continued*)

**Table 6.** (Continued)

| Variables | Status | | CHR(95%CI) | AHR (95%CI) |
|---|---|---|---|---|
| | Event | Censored | | |
| Yes | 74 | 67 | 1.9(1.4–2.7) | 1.4(0.96–2.2) |
| No | 78 | 187 | 1 | 1 |
| Hypothermia | | | | |
| Yes | 124 | 164 | 2.1(1.4–3.2) | 0.83(0.51–1.4) |
| No | 28 | 90 | 1 | 1 |
| Birth weight(grams) | | | | |
| < 1500 | 30 | 12 | 8.14(4.6–14.34) | **4.5(2.0–10.9)\*\*\*** |
| 1500–2499 | 102 | 81 | 5.16(3.19–8.32) | **3.1(1.4–6.6)\*\*** |
| ≥ 2500 | 20 | 161 | 1 | 1 |
| PNA | | | | |
| Yes | 50 | 49 | 2.04(1.5–2.87) | **2.7(1.8–4)\*\*\*** |
| No | 102 | 205 | 1 | 1 |
| Mode of delivery | | | | |
| CS | 32 | 80 | 0.73(0.49–1.1) | 0.7(0.46–1.1) |
| IA | 11 | 11 | 1.4(0.78–2.69) | 1.3(0.6–2.6) |
| SVD | 109 | 163 | 1 | 1 |

ANC = Antenatal Care, PROM = Premature rupture of membrane, RH = Rhesus factor, PNA = Perinatal asphyxia, CS = caesarean section, IA = Instrumental assisted, SVD = Spontaneous vaginal delivery, CHR = Crude Hazard Ratio, AHR = Adjusted Hazard Ratio

NB:

\*\*\* Significant (P-value < 0.001)

\*\* significant (p-value<0.005)

\* significant (p<0.05).

According to the multivariable Weibull regression model in our study very low birth weight and low birth weight were associated with the increment of mortality rate among neonates with RDS, which is supported by Studies conducted in Macedonia [29], Serbia [43], Fiji [46], Bangladesh [31], India [30], and China [36]. The reason for this finding is that due to low birth weight is a high-risk factor for physiological and anatomical immaturity in new-borns happen including a lack of surfactant due to lung immaturity [51].

In agreement with studies conducted in Serbia [43], Beijing China [36], and Cameron [33] PNA is a significant predictor and increases the hazard of death in the current study.. Hypoxia resulting from asphyxia at birth can directly damage alveolar type II epithelial cells and reduce surfactant production which will result in increased RDS incidence and mortality as well. In addition, the other justification is this be because of the following two reasons: Acute lung damage brought on by severe birth asphyxia reduces pulmonary surfactant production and secretion and also hypoxia reduces pulmonary surfactant activity and may even result in its inactivation [52].

This study revealed that Chorioaminoitis or maternal infection during the last pregnancy time or at the time of giving birth is a predictor and increases the hazards of death by two times among neonates with RDS. This finding is supported by a study conducted in China [36]. This is due to Chorioaminoitis is the leading cause of very preterm birth which the main cause of the occurrence of RDS and, therefore, contributes significantly to neonatal mortality [53]. The other reason is that the Chorioaminoitis decreases the synthesis, and inhibits the activity and secretion of Surfactants [54].

This study revealed multiple pregnancy as an independent predictor and increase the hazard of death by two times among new-born admitted with RDS and is negatively associated with neonatal survival with RDS. The reason is multiple pregnancies have a greater risk of preterm, low birth weight, organ immaturity, this is a likely reason to prone the neonates to lung immaturities and surfactant production reduction, which will increase the likelihood of mortality among neonates with RDS [55].

In this study, being born from mothers who administered antenatal corticosteroids before birth is a significant protective against death from RDS and those new-borns die 67% less likely than neonates born from mothers who didn't taken an antenatal corticosteroid. This finding is supported by different literature conducted in India [32], Iran [56] and in Fiji [46]. This is due to corticosteroids improve fetal lung development to prevent RDS after delivery. Steroid increases tissue and alveolar surfactant synthesis, influence the morphogenesis of the air spaces and functional differentiation of the embryonic lung. In addition, corticosteroids increase the expression of surfactant-associated proteins, decrease microvascular permeability, and speed up the structural development of the lungs overall. They also promote the generation of surfactant phospholipids by alveolar type II cells (through the fibroblast-pneumonocyte factor) [57–60].

## Limitations of the study

Since the design is based on secondary data, the study was unable to exhaustively explore all predictor variables that affect mortality. Variables like maternal nutritional status and neonatal vitamin D status were overlooked which greatly affects the outcome of RDS.

## Conclusion

From the study, the incidence of mortality from respiratory distress syndrome was found to be high. Perinatal asphyxia, multiple pregnancy, very low and low birth weight and Chorioaminoitis were statistically significant predictors of mortality when antenatal corticosteroid administration is a significant protective against mortality from RDS.

## Supporting information

**S1 Checklist. STROBE checklist.**
(DOCX)

**S1 File.**
(DTA)

## Acknowledgments

The authors acknowledge University of Gondar for securing ethical clearance for this study. The authors also want to acknowledge the five West Oromia Comprehensive Specialized Hospitals' administrations, data collectors, and supervisors for their respective unreserved efforts during this research work.

## Author Contributions

**Conceptualization:** Bruck Tesfaye Legesse, Tewodros Getaneh Alemu.

**Data curation:** Bruck Tesfaye Legesse.

**Formal analysis:** Bruck Tesfaye Legesse.

**Methodology:** Bruck Tesfaye Legesse, Tewodros Getaneh Alemu, Kendalem Asmare Atalell.

**Software:** Bruck Tesfaye Legesse.

**Validation:** Bruck Tesfaye Legesse, Kendalem Asmare Atalell.

**Visualization:** Bruck Tesfaye Legesse, Netsanet Melkamu Abera, Tewodros Getaneh Alemu, Kendalem Asmare Atalell.

**Writing – original draft:** Bruck Tesfaye Legesse.

**Writing – review & editing:** Netsanet Melkamu Abera, Tewodros Getaneh Alemu, Kendalem Asmare Atalell.

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
