## [Decision Letter · Decision Letter 0]

23 Jan 2023

PONE-D-22-27113Incidence and predictors of mortality among neonates with Respiratory Distress Syndrome admitted at West Oromia Referral Hospitals, Ethiopia, 2022. Multi-centred Institution Based Retrospective Follow-up Study.PLOS ONE

Dear Dr. Legesse,

Thank you for submitting your manuscript to PLOS ONE. After careful consideration, we feel that it has merit but does not fully meet PLOS ONE’s publication criteria as it currently stands. Therefore, we invite you to submit a revised version of the manuscript that addresses the points raised during the review process.

We look forward to receiving your revised manuscript.

Kind regards,

Dereje Haile, MPH/RH

Academic Editor

PLOS ONE

Journal Requirements:

Funding not applicable.

Additional Editor Comments:

Reviewer1

Dear authors, I was quite interested in reading your work. Respiratory distress syndrome is currently one of the most significant public health issues when it comes to ensuring newborn survival globally.

Here I have gathered a few general remarks regarding your manuscript:

- The Cox regression model is widely used in medical research and the shape and scale parameters of baseline hazard are not taken into account. Since the proportionate hazard assumption was met (0.5725), the parametric survival regression analysis is not necessary.

- Do you have any evidence that the bassline hazed distribution RDS existed before your study?

- Your manuscript is not written in Standard English

- When using the term "PROM" you do not give the standard definition used in your study. Since the definition in use can have a relevant influence on the survival of the neonates you should clarify this issue.

- In method section “(1.1- 1.4.1)". Please clarify!

- In the study setting section, you do not mention the numbers of health workers in their respective professions –which would have a significant impact on your results and neonatal survival.

- In the source population section “all neonates’ charts”. Are going to conclude for records?

- In the Sample size determination section “(sample size was 384.16)”. Please clarify!

- Also, the first objective of your study was the incidence of mortality. Is that possible to calculate sample size by considering the single population formula?

- In the same section, you stated “the double population proportion formula for the second objective”. Please clarify!

- In the data collection section, you stated “supervised by the principal investigator”. Please clarify!

- In data processing and analysis, you reported” Kaplan-Meier survival curve”, your title is the incidence of mortality, it is better to report Nelson Aalen cumulative hazard

- In the same section, you checked multicollinearity by a variance inflation factor. Please report the mean VIF or the cut point

- How do you measure perinatal asphyxia 3.2% of the neonates were delivered at home

- In your section " Incidence of mortality," you report the incidence of mortality in each hospital. In Nekemte Comprehensive Specialized Hospital 65.49 and 56.94 deaths / 1000neonates days in Mettu Karl Comprehensive Specialized Hospital. This disparity in the incidence of mortality needs a possible explanation and would greatly improve the relevance of your manuscript.

- In the model comparison section, you reported both graphs and tables. Please report either table or graphs

- In the section on predictors of mortality among neonates with respiratory distress syndrome, you stated that “the test of equality for survival distribution for different categories of variables was performed with Kaplan-Meier and the log-rank test.” Please clarify!

- In Table 6, please add the percentage of cell value

Only after thorough revisions of your manuscript, I would deem it eligible for PLOS ONE.

Reviewer 2

General Comment

The objective has been focused, and the necessary data has been generated, as well as appropriate data analysis and interpretation of the results. It reached conclusions that were supported by data. However, the language used in the submitted manuscript is insufficient, both in terms of clearly communicating what is intended and meeting scientific publication quality standards.

Review comments are provided here, and the authors may wish to consider improving the quality of the manuscript, particularly in terms of content organization, formatting, and language, to make the manuscript potentially publishable in PLOS.

The 'PLoS Guidelines to Authors' were to be followed by the authors in terms of content organization and formatting .Please take note that a few of the comments in the various sections below very clearly show this.

The manuscript needs extensive language editing( Especially on the discussion section); authors may want to enlist the help of a language editor. As part of this review, there has been extensive language editing done to the manuscript (please see below comments in each section)

Line 3: Titles should be written in sentence case (only the first word of the text, proper nouns, and genus names are capitalized)

Line 33: Change it “Version” to lowercase as “version”

Line 38: Would replace with “were included”

Line 40: Would edit “survival time” as “time”

Line 44: Would remove the extra space between comma and the word antenatal

Line 51: for the key words use comma instead of semicolon

Line 60: Would replace “neonate’s” with “neonatal”

Line 61: Would edit “30% to 40%” as “30 to 40%”

Line 67-68: Would edit “While accounting for 12.8% in Poland, 46.9% in Nigeria, and 49.5% of neonatal death in Ethiopia respectively” as “While accounting for 12.8% in, 46.9% in, and 49.5% of neonatal death in Poland, Nigeria and Ethiopia respectively ”.

Line 70: Would replace “Length of hospital stay” with “prolonged hospital stay,”

Line 82: Would edit “death outcomes” as “outcome”

Line 88: Use the acronym RDS; already used at first enter above with the expanded forms.

Line 90: Same as on line 88

Line 96-97: Same as on line 88

Line 103-104: Would edit “from May 01 to May 30, 2022.” as “on May, 2022.”

Line 115: would edit the sub heading “Study participants” as “Study population”

Line 120: Would you explain why you consider as exclusion criteria?

Line 124-127: Would edit as “The sample size was determined by using single population proportion formula for the first objective and double population proportion formula for the second objective. The following statistical assumptions were considered for single population proportion; death rate= 50% (p=0.5 and q=0.5), 95% confidence level, and 5% margin of error .

Line 130-131: Would edit as “Sampling frame was prepared after collecting the medical registration number of patients from the registration books of each hospital.”

Line 141: Would edit Sex, Place of birth” as “sex, place of birth”

Line 150: Would replace “Variables definition” with “Operational definitions”

Line 151-153: what is the value of operationalizing RDS? And also you already defined it on the introduction section.

Line 181: Would change “Preliminary” to lower case

Line 182-184: Would edit “Data collectors are nurses who have experience working in the NICU ward . In addition, data collectors were taken a one-day training from each hospital before actual data collection.” as “Data were collected by trained nurses having work experience in the NICU ward from respective hospitals”

Line 210-211: Would edit “A total of 406 neonates with respiratory distress syndrome medical records were participated in the study” as “A total of 406 RDS diagnosed neonates’ medical records were included in the study”

Table 1: Line 215-216: Use acronym RDS on the title

: remove “hr” from both “ ≤ 24hr and >24 hr”

: Would edit “35 and above” as “≥ 35”

Table 2: Line 222-223: share some comments as Table 1 and

: Legends shall not be part of the table, please put it on the footnotes

On table 3 and 4: share some comments as Table 1 and Table 2

Line 249-253: Would edit as “Models were compared graphically (Figure 2), and statistically by using Akaike information criteria (AIC) and log-likelihood to select the most parsimonious models for the data set.

Line 255: Would edit “The baseline hazard (the effect of time when all categorical variables are at a reference category) of experiencing an event/death was” as “The baseline hazard of death was . . . ”

**the word “experiencing” is not recommended for this manuscript outcome.

Line 260-261: same as previous

Line 263: use the acronym RDS

Line 264: remove “(death)”

Line 267-268: Would edit “The median follow-up survival time was 11 days (95% CI: 10-23).” as “The median time of follow-up was 11 days (95% CI: 10-23)”

Line 272-277: use acronyms for the name of each hospitals as on the previous

Line 290-291: Would edit “Furthermore, the log-rank test confirmed whether the observed difference was seen on the KM graph statistical difference or not.” as “Furthermore, the log-rank test confirmed whether the observed difference seen on the KM graph was statistically significant or not.”

Line 296: Would replace “those new-borns from mothers who developed Chorioaminoitis” as “their counterparts.”

Line 298: Would replace “new-borns” with “neonates”

Line 302: would edit “Bivariable and multivariable” as “Bi-variable and multi-variable”

Line 303: Would edited “Bi-variable” as “ bi-variable”

Line 311-315: Would avoid the statement “Keeping out other variables constant”, on multi-variable regression it can be known by default. And also Would suggest to interpret Hazard ratios like “at any given time the risk of death from RDS among neonates born from multiple pregnancies is twice that of neonates born from single.” Would do the same on this section

Line 326-328: would do the same as previous comments

Table 6: require the same correction as the previous tables

Line 335: Would edit “higher” as “much higher”

Line 336-338: Would edited as “This marked difference might be attributed to some factors such as the source population in the Fiji study were all live birth while the current study was exclusively on RDS”

Line 345-346: Would edit “study conducted in Bangladesh (36.5%) ” as “ in Bangladesh (36.5%)”

Line 364: Would edit “study design” as “study setting”

Line 364-369: Would edit as “The study in Kenya was only conducted among 92 neonates and mono-centric, while the current study is multi-centre. The other possible reason for this difference might be the study in Kenya was only within preterm and low birth weight which facilitates neonatal death while the current study includes all birth weight groups.”

Line 392-394: Would edit as “This study revealed multiple pregnancy as an independent predictor and increase the hazard of death by two times among new-born admitted with RDS.

Line 397: Would edit “RDS patients neonates” as “neonates with RDS.”

Line 421: Would remove “ETB: Ethiopian Birr”

Line 424: Would edit ”Pre Mature” as “Premature . . . “

Reviewers' comments:

Reviewer's Responses to Questions

**Comments to the Author**

1. Is the manuscript technically sound, and do the data support the conclusions?

Reviewer #1: Yes

Reviewer #2: Yes

2. Has the statistical analysis been performed appropriately and rigorously? 

Reviewer #1: Yes

Reviewer #2: Yes

3. Have the authors made all data underlying the findings in their manuscript fully available?

Reviewer #1: Yes

Reviewer #2: Yes

4. Is the manuscript presented in an intelligible fashion and written in standard English?

Reviewer #1: No

Reviewer #2: No

5. Review Comments to the Author

Reviewer #1: Dear authors, I was quite interested in reading your work. Respiratory distress syndrome is currently one of the most significant public health issues when it comes to ensuring newborn survival globally.

Here I have gathered a few general remarks regarding your manuscript:

- The Cox regression model is widely used in medical research and the shape and scale parameters of baseline hazard are not taken into account. Since the proportionate hazard assumption was met (0.5725), the parametric survival regression analysis is not necessary.

- Do you have any evidence that the bassline hazed distribution RDS existed before your study?

- Your manuscript is not written in Standard English

- When using the term "PROM" you do not give the standard definition used in your study. Since the definition in use can have a relevant influence on the survival of the neonates you should clarify this issue.

- In method section “(1.1- 1.4.1)". Please clarify!

- In the study setting section, you do not mention the numbers of health workers in their respective professions –which would have a significant impact on your results and neonatal survival.

- In the source population section “all neonates’ charts”. Are going to conclude for records?

- In the Sample size determination section “(sample size was 384.16)”. Please clarify!

- Also, the first objective of your study was the incidence of mortality. Is that possible to calculate sample size by considering the single population formula?

- In the same section, you stated “the double population proportion formula for the second objective”. Please clarify!

- In the data collection section, you stated “supervised by the principal investigator”. Please clarify!

- In data processing and analysis, you reported” Kaplan-Meier survival curve”, your title is the incidence of mortality, it is better to report Nelson Aalen cumulative hazard

- In the same section, you checked multicollinearity by a variance inflation factor. Please report the mean VIF or the cut point

- How do you measure perinatal asphyxia 3.2% of the neonates were delivered at home

- In your section " Incidence of mortality," you report the incidence of mortality in each hospital. In Nekemte Comprehensive Specialized Hospital 65.49 and 56.94 deaths / 1000neonates days in Mettu Karl Comprehensive Specialized Hospital. This disparity in the incidence of mortality needs a possible explanation and would greatly improve the relevance of your manuscript.

- In the model comparison section, you reported both graphs and tables. Please report either table or graphs

- In the section on predictors of mortality among neonates with respiratory distress syndrome, you stated that “the test of equality for survival distribution for different categories of variables was performed with Kaplan-Meier and the log-rank test.” Please clarify!

- In Table 6, please add the percentage of cell value

Only after thorough revisions of your manuscript, I would deem it eligible for PLOS ONE.

Reviewer #2: General Comment

The objective has been focused, and the necessary data has been generated, as well as appropriate data analysis and interpretation of the results. It reached conclusions that were supported by data. However, the language used in the submitted manuscript is insufficient, both in terms of clearly communicating what is intended and meeting scientific publication quality standards.

Review comments are provided here, and the authors may wish to consider improving the quality of the manuscript, particularly in terms of content organization, formatting, and language, to make the manuscript potentially publishable in PLOS.

The 'PLoS Guidelines to Authors' were to be followed by the authors in terms of content organization and formatting .Please take note that a few of the comments in the various sections below very clearly show this.

The manuscript needs extensive language editing( Especially on the discussion section); authors may want to enlist the help of a language editor. As part of this review, there has been extensive language editing done to the manuscript (please see below comments in each section)

Line 3: Titles should be written in sentence case (only the first word of the text, proper nouns, and genus names are capitalized)

Line 33: Change it “Version” to lowercase as “version”

Line 38: Would replace with “were included”

Line 40: Would edit “survival time” as “time”

Line 44: Would remove the extra space between comma and the word antenatal

Line 51: for the key words use comma instead of semicolon

Line 60: Would replace “neonate’s” with “neonatal”

Line 61: Would edit “30% to 40%” as “30 to 40%”

Line 67-68: Would edit “While accounting for 12.8% in Poland, 46.9% in Nigeria, and 49.5% of neonatal death in Ethiopia respectively” as “While accounting for 12.8% in, 46.9% in, and 49.5% of neonatal death in Poland, Nigeria and Ethiopia respectively ”.

Line 70: Would replace “Length of hospital stay” with “prolonged hospital stay,”

Line 82: Would edit “death outcomes” as “outcome”

Line 88: Use the acronym RDS; already used at first enter above with the expanded forms.

Line 90: Same as on line 88

Line 96-97: Same as on line 88

Line 103-104: Would edit “from May 01 to May 30, 2022.” as “on May, 2022.”

Line 115: would edit the sub heading “Study participants” as “Study population”

Line 120: Would you explain why you consider as exclusion criteria?

Line 124-127: Would edit as “The sample size was determined by using single population proportion formula for the first objective and double population proportion formula for the second objective. The following statistical assumptions were considered for single population proportion; death rate= 50% (p=0.5 and q=0.5), 95% confidence level, and 5% margin of error .

Line 130-131: Would edit as “Sampling frame was prepared after collecting the medical registration number of patients from the registration books of each hospital.”

Line 141: Would edit Sex, Place of birth” as “sex, place of birth”

Line 150: Would replace “Variables definition” with “Operational definitions”

Line 151-153: what is the value of operationalizing RDS? And also you already defined it on the introduction section.

Line 181: Would change “Preliminary” to lower case

Line 182-184: Would edit “Data collectors are nurses who have experience working in the NICU ward . In addition, data collectors were taken a one-day training from each hospital before actual data collection.” as “Data were collected by trained nurses having work experience in the NICU ward from respective hospitals”

Line 210-211: Would edit “A total of 406 neonates with respiratory distress syndrome medical records were participated in the study” as “A total of 406 RDS diagnosed neonates’ medical records were included in the study”

Table 1: Line 215-216: Use acronym RDS on the title

: remove “hr” from both “ ≤ 24hr and >24 hr”

: Would edit “35 and above” as “≥ 35”

Table 2: Line 222-223: share some comments as Table 1 and

: Legends shall not be part of the table, please put it on the footnotes

On table 3 and 4: share some comments as Table 1 and Table 2

Line 249-253: Would edit as “Models were compared graphically (Figure 2), and statistically by using Akaike information criteria (AIC) and log-likelihood to select the most parsimonious models for the data set.

Line 255: Would edit “The baseline hazard (the effect of time when all categorical variables are at a reference category) of experiencing an event/death was” as “The baseline hazard of death was . . . ”

**the word “experiencing” is not recommended for this manuscript outcome.

Line 260-261: same as previous

Line 263: use the acronym RDS

Line 264: remove “(death)”

Line 267-268: Would edit “The median follow-up survival time was 11 days (95% CI: 10-23).” as “The median time of follow-up was 11 days (95% CI: 10-23)”

Line 272-277: use acronyms for the name of each hospitals as on the previous

Line 290-291: Would edit “Furthermore, the log-rank test confirmed whether the observed difference was seen on the KM graph statistical difference or not.” as “Furthermore, the log-rank test confirmed whether the observed difference seen on the KM graph was statistically significant or not.”

Line 296: Would replace “those new-borns from mothers who developed Chorioaminoitis” as “their counterparts.”

Line 298: Would replace “new-borns” with “neonates”

Line 302: would edit “Bivariable and multivariable” as “Bi-variable and multi-variable”

Line 303: Would edited “Bi-variable” as “ bi-variable”

Line 311-315: Would avoid the statement “Keeping out other variables constant”, on multi-variable regression it can be known by default. And also Would suggest to interpret Hazard ratios like “at any given time the risk of death from RDS among neonates born from multiple pregnancies is twice that of neonates born from single.” Would do the same on this section

Line 326-328: would do the same as previous comments

Table 6: require the same correction as the previous tables

Line 335: Would edit “higher” as “much higher”

Line 336-338: Would edited as “This marked difference might be attributed to some factors such as the source population in the Fiji study were all live birth while the current study was exclusively on RDS”

Line 345-346: Would edit “study conducted in Bangladesh (36.5%) ” as “ in Bangladesh (36.5%)”

Line 364: Would edit “study design” as “study setting”

Line 364-369: Would edit as “The study in Kenya was only conducted among 92 neonates and mono-centric, while the current study is multi-centre. The other possible reason for this difference might be the study in Kenya was only within preterm and low birth weight which facilitates neonatal death while the current study includes all birth weight groups.”

Line 392-394: Would edit as “This study revealed multiple pregnancy as an independent predictor and increase the hazard of death by two times among new-born admitted with RDS.

Line 397: Would edit “RDS patients neonates” as “neonates with RDS.”

Line 421: Would remove “ETB: Ethiopian Birr”

Line 424: Would edit ”Pre Mature” as “Premature . . . “

6. PLOS authors have the option to publish the peer review history of their article (what does this mean?). If published, this will include your full peer review and any attached files.

Reviewer #1: No

Reviewer #2: No

---

## [Author Response · Author response to Decision Letter 0]

16 Mar 2023

Thank you very much for your precious comments, questions, and suggestions.

We tried to answer for all of the raised concerns with letter called responses for the reviewers separably.

---

## [Decision Letter · Decision Letter 1]

10 May 2023

PONE-D-22-27113R1Incidence and predictors of mortality among Neonates with Respiratory Distress Syndrome admitted at West Oromia Referral Hospitals, Ethiopia, 2022. Multi-centred Institution Based Retrospective Follow-up Study.PLOS ONE

Dear Dr. Legesse,

Thank you for submitting your manuscript to PLOS ONE. After careful consideration, we feel that it has merit but does not fully meet PLOS ONE’s publication criteria as it currently stands. Therefore, we invite you to submit a revised version of the manuscript that addresses the points raised during the review process.

We look forward to receiving your revised manuscript.

Kind regards,

Dereje Haile, MPH/RH

Academic Editor

PLOS ONE

Journal Requirements:

Additional Editor Comments:

Reviewer 1

Thank you for revising your manuscript in response to my comments.

Please check the grammar. For example, in line #91, after "however," commas are needed! Please read the whole document.

There is an inconsistency in the method section "1.1–1.4; please remove it.

You calculated the sample size using the conventional method, or 50%. However, a study conducted at Gondar hospital reported that the probability of mortality among RDS children was 16.37. Hence, I recommend that you use the sample size method for the Cox model or the Freedman method by using the following Stata command:

power cox, failprob (.1637) wdprob (0.05)

Reviewers' comments:

Reviewer's Responses to Questions

**Comments to the Author**

1. If the authors have adequately addressed your comments raised in a previous round of review and you feel that this manuscript is now acceptable for publication, you may indicate that here to bypass the “Comments to the Author” section, enter your conflict of interest statement in the “Confidential to Editor” section, and submit your "Accept" recommendation.

Reviewer #1: All comments have been addressed

2. Is the manuscript technically sound, and do the data support the conclusions?

Reviewer #1: Yes

3. Has the statistical analysis been performed appropriately and rigorously? 

Reviewer #1: Yes

4. Have the authors made all data underlying the findings in their manuscript fully available?

Reviewer #1: Yes

5. Is the manuscript presented in an intelligible fashion and written in standard English?

Reviewer #1: No

6. Review Comments to the Author

Reviewer #1: Thank you for revising your manuscript in response to my comments.

Please check the grammar. For example, in line #91, after "however," commas are needed! Please read the whole document.

There is an inconsistency in the method section "1.1–1.4; please remove it.

You calculated the sample size using the conventional method, or 50%. However, a study conducted at Gondar hospital reported that the probability of mortality among RDS children was 16.37. Hence, I recommend that you use the sample size method for the Cox model or the Freedman method by using the following Stata command:

power cox, failprob (.1637) wdprob (0.05)

7. PLOS authors have the option to publish the peer review history of their article (what does this mean?). If published, this will include your full peer review and any attached files.

Reviewer #1: No

---

## [Author Response · Author response to Decision Letter 1]

24 Jun 2023

We are thankful for the constructive comments and suggestions.

---

## [Editor Report · Decision Letter 2]

11 Jul 2023

Incidence and predictors of mortality among Neonates with Respiratory Distress Syndrome admitted at West Oromia Referral Hospitals, Ethiopia, 2022. Multi-centred Institution Based Retrospective Follow-up Study.

PONE-D-22-27113R2

Dear Dr. Bruck Tesfaye Legesse,

We’re pleased to inform you that your manuscript has been judged scientifically suitable for publication and will be formally accepted for publication once it meets all outstanding technical requirements.

Kind regards,

Dereje Haile, MPH/RH

Academic Editor

PLOS ONE
---

## [Editor Report · Acceptance letter]

21 Jul 2023

PONE-D-22-27113R2 

Incidence and predictors of mortality among Neonates with Respiratory Distress Syndrome admitted at West Oromia Referral Hospitals, Ethiopia, 2022. Multi-centred Institution based retrospective follow-up study. 

Dear Dr. Legesse:

I'm pleased to inform you that your manuscript has been deemed suitable for publication in PLOS ONE. Congratulations! Your manuscript is now with our production department. 

Kind regards, 

on behalf of

Mr. Dereje Haile 

Academic Editor

PLOS ONE